Looking for the sponge loop: analyses of detritus on a Caribbean forereef using stable isotope and eDNA metabarcoding techniques

Olinger Lauren K. l.olinger12993@gmail.com 1 2
McClenaghan Beverly 3
Hajibabaei Mehrdad 3 4
Fahner Nicole 3
Berghuis Lesley 3
Rajabi Hoda 3
Erwin Patrick 2
Lane Chad S. 5
Pawlik Joseph R. 2
1 Center for Marine and Environmental Studies, University of the Virgin Islands , St Thomas, Virgin Islands , U.S. Virgin Islands , United States of America
2 Department of Biology and Marine Biology, University of North Carolina Wilmington , Wilmington , NC , United States of America
3 eDNAtec Inc. , Newfoundland and Labrador , St. John’s , Canada
4 Department of Integrative Biology, University of Guelph , Guelph , Ontario , Canada
5 Department of Earth and Ocean Sciences, University of North Carolina Wilmington , Wilmington , NC , United States of America
Johnson Magnus
Electronic publication date: 2024 Feb 23
Publication date: 2024
Volume: 12
Electronic Location ID: e16970
Received 2023 Nov 27; Accepted 2024 Jan 29
Copyright: ©2024 Olinger et al.
Copyright year: 2024
Copyright holder: Olinger et al.
License: This is an open access article distributed under the terms of the Creative Commons Attribution License, which permits unrestricted use, distribution, reproduction and adaptation in any medium and for any purpose provided that it is properly attributed. For attribution, the original author(s), title, publication source (PeerJ) and either DOI or URL of the article must be cited.
License URL: https://creativecommons.org/licenses/by/4.0/

Keywords: Detritus, Coral reef, Trophodynamics, Porifera, eDNA, Stable isotope analyses

Funding: National Science Foundation OCE 2218863 Atlantic Canada Opportunities Agency’s Atlantic Innovation Fund project number 781-37749-207993 Energy Research and Innovation Newfoundland and Labrador (ERINL) Any opinions, findings, and conclusions or recommendations expressed in this publication are those of the authors and do not necessarily reflect the views of ERINL or its members This work was funded by the National Science Foundation (OCE 2218863), the Atlantic Canada Opportunities Agency’s Atlantic Innovation Fund (project number 781-37749-207993), and a grant from Energy Research and Innovation Newfoundland and Labrador (ERINL). Any opinions, findings, and conclusions or recommendations expressed in this publication are those of the authors and do not necessarily reflect the views of ERINL or its members. There was no additional external funding received for this study. The funders had no role in study design, data collection and analysis, decision to publish, or preparation of the manuscript.

==============================
Coral reefs are biodiverse ecosystems that rely on trophodynamic transfers from primary producers to consumers through the detrital pathway. The sponge loop hypothesis proposes that sponges consume dissolved organic carbon (DOC) and produce large quantities of detritus on coral reefs, with this turn-over approaching the daily gross primary production of the reef ecosystem. In this study, we collected samples of detritus in the epilithic algal matrix (EAM) and samples from potential sources of detritus over two seasons from the forereef at Carrie Bow Cay, Belize. We chose this location to maximize the likelihood of finding support for the sponge loop hypothesis because Caribbean reefs have higher sponge abundances than other tropical reefs worldwide and the Mesoamerican barrier reef is an archetypal coral reef ecosystem. We used stable isotope analyses and eDNA metabarcoding to determine the composition of the detritus. We determined that the EAM detritus was derived from a variety of benthic and pelagic sources, with primary producers (micro- and macroalgae) as major contributors and metazoans (Arthropoda, Porifera, Cnidaria, Mollusca) as minor contributors. None of the sponge species that reportedly produce detritus were present in EAM detritus. The cnidarian signature in EAM detritus was dominated by octocorals, with a scarcity of hard corals. The composition of detritus also varied seasonally. The negligible contribution of sponges to reef detritus contrasts with the detrital pathway originally proposed in the sponge loop hypothesis. The findings indicate a mix of pelagic and benthic sources in the calmer summer and primarily benthic sources in the more turbulent spring.

Introduction

Coral reefs are among the most biodiverse marine ecosystems despite occurring in oligotrophic tropical waters (Darwin & Bonney, 1896). The paradoxical survival of coral reefs in such marine deserts is made possible by high primary productivity and efficient nutrient cycles (Mumby & Steneck, 2018). However, the survival of coral reefs is at risk due to human-driven climate change, diseases, overfishing, and other stressors (Hughes, 1994; Hoegh-Guldberg et al., 2007). These stressors are driving the replacement of hermatypic corals by non-reef building organisms, such as algae, sponges, and benthic cyanobacterial mats (BCM), reducing reef complexity, accretion, water quality, and overall habitat value for higher trophic levels (Done, 1992; Aronson et al., 2002; Fabricius, 2005; Norström et al., 2009; De Bakker et al., 2017). Changes to benthic communities also cause cascading effects to the nutrient cycles on which these ecosystems rely, including how energy is transferred from primary producers to higher trophic levels through the detrital pathway (Begon, Harper & Townsend, 1986; Johnson et al., 1995; Wilson et al., 2003).

Detritus is a protein- and amino acid-rich mixture (Wilson, 2000; Crossman et al., 2001) of non-living organic matter derived from non-fossil living sources and living organic matter from heterotrophic and autotrophic microbes (Begon, Harper & Townsend, 1986; Wilson et al., 2003). The combination of detritus and inorganic carbonates and silicates forms benthic particulates on coral reefs (Tebbett & Bellwood, 2019). The accumulation of detritus-laden particulates in the turf-forming algae that covers dead coral surfaces is collectively known as the epilithic algal matrix (EAM). This amalgamation of nutritious detritus and algae is an important food resource for many fish and invertebrate detritivores (Hatcher, 1983; Purcell & Bellwood, 2001; Wilson et al., 2003; Kramer, Bellwood & Bellwood, 2012).

Detrital organic matter on coral reefs is derived from a variety of autochthonous and allochthonous benthic and pelagic sources. Micro- and macro-algae are major sources of detritus that accumulate on the benthos following mechanical breakdown by physical forces or through grazing (Hatcher, 1983; Alongi, 1988; Wilson et al., 2003). Algae can also produce detritus indirectly by generating photosynthate, forms of dissolved organic carbon (DOC) that can aggregate into non-cellular amorphic detritus by adsorbing to carbonate sediments or spontaneously forming polymer gels (Otsuki & Wetzel, 1973; Chin, Orellana & Verdugo, 1998; Wilson et al., 2003). Another source of detritus is bacteria (Tebbett & Bellwood, 2019) that can form biofilms and act as nucleation sites for DOC aggregation, and heterotrophic bacteria can consume detritus and mediate its transformation to a form that can be consumed by metazoans (Biddanda, 1985; Alongi, 1988; Wilson et al., 2003). Benthic cyanobacterial mats (BCM) are also becoming more abundant on Caribbean coral reefs (De Bakker et al., 2017) and may be similar to benthic macroalgae with respect to their contribution to detritus, either directly through mechanical breakdown following physical detachment from the reef, or indirectly through production of large quantities DOC (Brocke et al., 2015) that may aggregate and build up in the EAM.

Detritus may also be derived from the shedding of mucus or cells by benthic metazoans, namely corals and sponges. Coral mucus has long been considered an important source of detritus on coral reefs (Gottfried & Roman, 1983). Sponges have more recently been implicated in detritus production, and sponge conversion of DOC to detritus may be an important mechanism recycling carbon back to the benthos before it can escape the reef, thereby aiding in tight carbon cycles that ensure reef survival under oligotrophic conditions (De Goeij et al., 2013). This conversion of DOC to detritus is the theorized sponge loop, named after the similar microbial loop which describes the return of DOC back into the grazing food web via predation on free-living microbes by protists (Azam et al., 1983). Since its proposal, the sponge loop hypothesis has been widely cited and has been featured in undergraduate marine biology textbooks (Levinton, 2021).

The sponge loop hypothesis was originally proposed based on incubation experiments using flow chambers or open pots spiked with isotopically enriched DOC containing thinly encrusting, cryptic (found in reef interstices) Caribbean reef sponge species, with the isotopic signature traced in detritus collected by filtration or by pipette (De Goeij et al., 2013). Four cryptic sponge species were used in these experiments (Halisarca caerulea, Haliclona vansoesti, Chondrilla caribensis, and Scopalina ruetzleri) with the mechanism of detritus production proposed as rapid cellular proliferation and choanocyte turnover associated with space-limitation and restricted sponge growth (De Goeij et al., 2013). Detritus production via the sponge loop was proposed to be very large, with sponges (both cryptic and massive, or emergent, species) transforming most of the DOC they consumed into detritus, estimated at 9.3% of sponge biomass per day (Alexander et al., 2014) and with the turnover of DOC to detritus approaching the daily gross primary production of the entire reef ecosystem (De Goeij et al., 2013). However, subsequent experiments with larger, emergent sponge species that grow on the reef surface failed to find these sponges to be net producers of detritus and instead suggested that they return DOC to the benthos as sponge biomass (McMurray et al., 2018). More recently, some emergent sponge species, including the Caribbean species Aplysina archeri, were observed “sneezing” mucus and particulates from near their incurrent ostia (Kornder et al., 2022), but it is not yet clear if this is a method of detritus production analogous to that of cryptic sponge species or whether this mechanism is common across the emergent sponge fauna. The discovery of the microbial loop accounted for a large fraction of missing oceanic carbon (Fenchel, 2008), but it is unclear whether the carbon transformation mediated by the sponge loop is as important a pathway.

Detritus collected from a forereef off Palmyra Atoll showed abundant pelagic inputs (Max et al., 2013). Studies such as those conducted on the Palmyra Atoll by Max et al. (2013) used stable isotope analyses (SIA) to determine the sources contributing to benthic detritus. SIA is a popular technique for tracing energy flow through ecosystems, and it is based on the discrimination against heavier isotopes in metabolic processes, which separates organisms from different trophic levels along axes of 15N/14N and separates organisms with different types of primary productivity and carbon sources along axes of 13C/12C (Post, 2002; Michener & Kaufman, 2007; Middelburg, 2014; Zapata-Hernández et al., 2021). SIA has been relatively underused to explore trophodynamics on coral reefs (Greenwood, Sweeting & Polunin, 2010).

While popular, SIA is not able to discern between species with similar trophic levels and carbon-fixing pathways. This limitation can be resolved with supplemental collection and metabarcoding of environmental DNA (eDNA) samples, an increasingly popular approach for characterizing biodiversity that does not require collection of whole biological specimens and instead relies on recovery and analysis of DNA from the physical environment in which they live (e.g., water, soil, sediment; Thomsen & Willerslev, 2015). This eDNA is released from organisms through various mechanisms, including cell shedding and the excretion of various bodily fluids and feces, and it can be isolated, amplified, and sequenced using high-throughput genomic sequencing platforms and then screened against publicly available databases (e.g., GenBank) to assign taxonomy to the organisms contributing to that environmental sample (Baird & Hajibabaei, 2012; Taberlet et al., 2012; Thomsen & Willerslev, 2015). Analyses of eDNA have been used to quantify biodiversity (Thomsen & Willerslev, 2015), determine species diets (Boyi et al., 2022), and identify elusive species (McClenaghan et al., 2020). Analyses of eDNA can also complement SIA-based estimations of source contributions to organic matter resources, such as detritus. A similar SIA/eDNA approach was recently used to quantify the relative contributions of different sources to sedimentary organic carbon in seagrass meadows (Reef et al., 2017).

In this study, we chose to test the sponge loop hypothesis from the “sink side”, focusing on the sponge signature in natural EAM detritus accumulating on a coral reef, rather than from the “source side” as detritus is produced by targeted sponges, whether in laboratory incubations or in the field (De Goeij et al., 2013; McMurray et al., 2018, reviewed in Pawlik & McMurray, 2020). Further, we chose a location for our study that would maximize the likelihood of confirming the sponge loop hypothesis in providing high sponge abundance. Caribbean reefs are distinct from others in the tropics in having higher sponge abundance (Wilkinson & Cheshire, 1990), with mean percentage cover of sponges on Caribbean forereefs of ∼16% (Loh & Pawlik, 2014). If, as proposed by the sponge loop hypothesis, >9% of sponge biomass per day is returned to the benthos as detritus and sponges cover >10% of the reef surface, we would anticipate that the sponge signal in natural EAM detritus would be very large. We collected samples of reef EAM detritus and samples from potential sources of detritus, including from organisms and from areas where detritus may originate (e.g., water column, fish feces) from a Caribbean forereef within the archetypal Mesoamerican barrier reef ecosystem (Carrie Bow Cay, Belize) over two seasons (July 2018 and March 2019). Stable isotope and eDNA metabarcoding analyses were applied to the samples to explore the composition of detritus, with a particular focus on sponges to test claims that sponges are major sources of detritus on Caribbean coral reefs (De Goeij et al., 2013). In addition to testing the sponge loop hypothesis, this study provided an opportunity to further develop both SIA- and eDNA-based methodologies, allowing comparisons across sampling seasons for the same reef location and comparisons with other studies.

Materials & Methods

Sample collection

Samples were collected in July 2018 and March 2019 from the forereef near the Smithsonian field station at Carrie Bow Cay, Belize. Carrie Bow Cay is situated on the Mesoamerican Barrier Reef, approximately 22 km from the city of Dangriga on mainland Belize. The characteristic spur-and-groove formation of the forereef at 15–20 m depth is an Orbicella annularis framework with abundant octocorals, sponges, and algae, and with hard coral cover estimated at ∼10% at the times of this study. The forereef location (16°48′10.6″N, 88°4′43.9″W) approximately 500 m northeast of the island of Carrie Bow Cay drops off steeply at the outer ridge less than 100 m to the east. Cuts to the north and south of the island and reef isolate the complex from the rest of the Mesoamerican barrier reef and expose the area to currents that flow through the channels, including waters from the open ocean and from the shoreward lagoon with large mangrove islands and seagrass beds nearby. The reef complex at Carrie Bow Cay has been extensively studied since its founding as a research site by scientists in 1972 (Rützler & Macintyre, 1982). Field experiments were approved by the Belize Fisheries Department (permit number 00023-18).

Two categories of samples were collected from the forereef: detritus and detritus sources, where the latter refers to organisms or materials that may be contributing to detritus. Samples also encompassed either tissues or composites, where tissues refer to organic material deliberately collected from a living organism and composites refer to organic remnants of multiple organisms that had been previously deposited to the water column or benthos as particulates or feces.

Two types of detritus were collected: detritus from the epilithic algal matrix (EAM detritus) and detritus from plastic trays placed under reef overhangs (tray detritus), with the latter added for the second sampling season. Seven types of source samples were collected: algae tissue, benthic cyanobacterial mat (BCM) tissue, feces from two species of herbivorous fish, feces from two species of spongivorous fish, tissues of three emergent sponge species, tissues of three cryptic sponge species, and water column particulates from sediment traps (Table 1) The difficulty in collecting some source samples (e.g., spongivores feces) is reflected in lower levels of sample replication.

Table 1 Number of replicates of detritus and detritus sources collected in July 2018 and March 2019 and analyzed using stable isotope analyses (nSIA) and genetic sequencing (nseq).

Category	Sample type	Species	July 2018	March 2019	
			n SIA	n seq	n SIA	n seq	
Detritus	EAM	Total	5	5	11	6	
Tray	Total	0	0	3	2	
Source	Algae tissue	Dictyota sp.	5	5	5	0	
		Halimeda sp.	0	0	3	0	
		Lobophora variegata	5	5	6	0	
	Total	10	10	14	0	
	BCM tissue	Total	3	3	5	0	
	Herbivore feces	Acanthurus bahianus	3	3	4	6	
		Acanthurus coeruleus	2	2	1	1	
	Total	5	5	5	7	
	Spongivore feces	Holacanthus ciliaris	1	1	1	2	
		Pomacanthus paru	0	0	2	2	
	Total	1	1	3	4	
	Emergent sponge tissue	Aplysina cauliformis	10	0	10	0	
		Niphates digitalis	8	0	10	0	
		Xestospongia muta	10	5	10	0	
	Total	28	5	30	0	
	Cryptic sponge tissue	Chondrilla sp.	2	2	3	0	
		Halisarca caerulea	3	3	4	0	
		Scopalina ruetzleri	3	3	5	0	
	Total	8	8	12	0	
Sediment trap	Total	5	5	14	7	
Notes.

Composite samples are indicated in bold.

EAM and tray detritus

Detrital material was suctioned from a 25 cm2 area of epilithic algal matrix (EAM) using 50 ml polypropylene syringes. EAM samples (2018 n = 5, 2019 n = 11) were collected at least 10 m apart and from flat patches of dead rock covered with closely cropped turf and devoid of pits (Purcell & Bellwood, 2001). In 2019, trays (white plastic bins with dimensions 40 × 31.8 × 15.2 cm; n = 6) were deployed under reef overhangs to target detrital outfall from organisms, including cryptic sponges. After 72 h, the particulates and detritus that had accumulated in the trays were suctioned up using 50 ml polypropylene syringes. Syringes of EAM and tray detritus were transported back to the lab at Carrie Bow Cay, where their contents were filtered through a 150 µm mesh to isolate the smaller size fraction targeted by detritivores (Wilson et al., 2003). The mixture was divided, with approximately 5 ml transferred to a microcentrifuge tube with 0.5 ml ethanol and the remaining filtered onto a 0.7 µm filter (Watson GFF), and both samples were stored at −20 °C.

Tissue samples

The tissues of organisms that are abundant and possible contributors to reef detritus were collected at the same location and time (July 2018, March 2019) as detritus collections. Divers sought out different individuals or patches of each targeted organism; species were visually identified on site, and identifications were later validated with genetic sequence data for all but the most conspicuous organisms (Aplysina cauliformis, Niphates digitalis, and Halimeda sp). When collecting the tissue samples, the sample quantity and collection method varied by species. The algae and benthic cyanobacterial mat (BCM) tissues were collected in handfuls, using snips to separate algae or bacterial tissues from each source patch, and transferred to the lab at Carrie Bow Cay in plastic bags. The tissues of emergent sponge species Xestospongia muta, A. cauliformis, and N. digitalis were collected with a dive knife or cork borer (one cm diameter), and clean pieces of tissue (approximately eight cm3 or two cm cylinders) were transported to the lab at Carrie Bow Cay in plastic bags. Tissues of cryptic sponge species Chondrilla sp., Halisarca caerulea, and Scopalina ruetzleri were removed from rock using a scalpel blade, producing tissue samples that were approximately four cm2 and 1–2 mm thick. All tissue samples were sealed at depth in plastic bags and transported back to the lab, where a subsample of tissue (<1 cm3) from each organism was transferred to a microcentrifuge tube, rinsed twice with ethanol, and stored in 0.5 ml ethanol at −20 °C. The plastic bag with the remaining tissue sample was squeezed of excess water, resealed, and frozen at −20 °C.

Feces samples

Feces of two herbivorous fish species (Acanthurus bahianus, Acanthurus coeruleus) and two spongivorous fish species (Holacanthus ciliaris, Pomacanthus paru) were also collected. For these samples, divers observed fish, and when the fish was seen defecating, the diver used a turkey baster to collect the feces before it touched the benthos. The contents of the turkey baster were transferred to a plastic bag, sealed at depth, and transported back to the lab. The mixture was divided, with approximately 5 ml transferred to a microcentrifuge tube with 0.5 ml ethanol and the remaining filtered onto a 0.7 µm filter (Watson GFF), and both samples were stored at −20 °C.

Sediment trap samples

Sediment traps made of PVC pipe with diameter of 7.6 cm and length of 30.5 cm (4:1 aspect ratio) were deployed on the reef for 72 h to capture water column particulates that may be contributing to detritus. While seawater samples would have been a more representative sample of pelagic contributions to reef detritus, this was not feasible due to the volume of water (> >4 L) required for stable isotope analyses. Traps (2018 n = 5; 2019 n = 15; Table 1) were deployed at least 10 m apart on the reef and affixed with cable ties to a metal stake that had been securely driven into dead coral heads. After 72 h, it was determined that sufficient sediment had accumulated for analyses, and plastic bags and rubber bands were used to seal sediment traps at depth before being transported to the lab at Carrie Bow Cay for processing. The mixture was divided, with approximately 5 ml transferred to a microcentrifuge tube with 0.5 ml ethanol and the remaining filtered onto a 0.7 µm filter (Watson GFF), and both samples were stored at −20 °C.

Stable isotope analyses

Samples were transported on ice to University of North Carolina Wilmington (UNCW), where tissue samples and filters containing composite samples were transferred to combusted scintillation vials and lyophilized for 24 h. Dried tissue samples were ground into a powder using a mortar and pestle, and the powder was returned to the scintillation vials. Filters were cut in half using a scalpel. One half of each filter and approximately 10 mg of each tissue sample was transferred to a second combusted vials for acidification by exposure to vapors of 12 M hydrochloric acid (HCl) for 24 h. Acidified samples were then placed on a 60 °C hotplate for 48 h to boil off remaining HCl. Each of the non-acidified and acidified filter halves were halved again, and the quarter filter segments were transferred to foil capsules. For each non-acidified and acidified tissue sample, an ultra-microbalance was used to precisely measure (within 5%) a target weight of sample into foil capsules. Target weights varied from 2 mg for Dictyota to 7 mg for Lobophora, and 4 mg for all other sampled species (six sponge species, BCM, and Halimeda).

Non-acidified and acidified samples were analyzed for nitrogen and carbon stable isotope compositions, respectively, using a Costech 4010 Elemental Analyzer interfaced with a Thermo Delta V Plus stable isotope mass spectrometer at UNCW. Isotopic results are expressed in standard delta (δ) units calculated as follows: δ13C or δ15N (‰) = 1000[(Rsample/Rstandard) − 1], where Rsample and Rstandard are the 13C/12C or 15N/14N ratios of samples and standards, respectively. Within-run standard deviations of δ13C and δ15N were < 0.3‰ across repeated analyses of USGS 40 and USGS 41a glutamic acid standards (average = 0.2‰, min = 0.1‰). A total of 8 samples from 2019 were omitted from analyses due to insufficient acidification leading to remnant inorganic carbon species, as determined by δ13C values that exceeded −10 (Schidlowski, 2001). For fish feces samples collected in 2019, the stable isotope composition of three feces samples (n = 2 from herbivore Acanthurus bahianus and n = 1 from spongivore Holocanthus ciliarus) were not measured due to insufficient material on the filter.

Metabarcoding analyses

The samples stored in microcentrifuge tubes in ethanol were shipped to the Center for Environmental Genomics Analysis (CEGA) for sequencing in 2019. CEGA designed and optimized a fully custom eDNA analysis. DNA extraction methods were optimized for each sample type to maximize DNA recovery, and DNA was extracted from all samples using the optimized protocols. Three DNA markers partially covering three genetic regions (cytochrome c oxidase I, 18S rRNA, 16S rRNA) were amplified from each sample using PCR (Tables S1–S2). These markers were chosen to provide comprehensive detection of metazoans, algae, and cyanobacteria. Negative controls were added during extraction and PCR steps and carried through to sequencing. The three amplicons from each sample were sequenced on an Illumina NovaSeq 6000 instrument with a target minimum sequencing depth of 250,000 sequences per sample per marker. See Supplemental Information for detailed DNA extraction and library preparation procedures. Raw sequence reads are available in NCBI’s sequence read archive under project PRJNA965826.

After DNA sequencing, base calling and demultiplexing were performed using Illumina’s bcl2fastq software (v2.20), and primers were trimmed from sequences using cutadapt (v2.8; Martin, 2011). Using DADA2 (v1.14; Callahan et al., 2016) with default parameters, sequences were filtered for quality and length and then denoised to create exact sequence variants (ESVs), each of which represent a unique sequence from the sample. ESVs were assigned taxonomy using NCBIs’s BLASTN tool (v2.11; Altschul et al., 1990) and the NT database (downloaded November 25, 2020) with an e-value cut-off of 0.001. The taxonomy presented here matches the naming conventions used in NCBI’s taxonomy database and was assigned based on a sequence similarity score (the product of the percent sequence similarity and the percent query coverage). The minimum scores for taxonomic assignment at each level were as follows: phylum at 85%, class and order at 90%, family at 95%, genus at 98%, and species at 99%. The results from all markers were consolidated to create the taxonomic lists. Taxonomic identifications were verified against the World Register of Marine Species (WoRMs), Global Biodiversity Information Facility (GBIF), and the Encyclopedia of Life (EOL) to ensure that spurious matches resulting from poor reference database coverage were not included in the list. As standard practice, ESVs assigned to humans and other common contaminants were removed from the sample data. Sequencing results from the negative controls generated during lab processing were screened for contamination, and any ESVs detected in the negative controls were removed from the associated samples. No ESVs were amplified from negative controls that were also detected in associated samples and could be assigned taxonomy based on the selection criteria described above.

Data analyses

Bayesian stable isotope mixing models (R package SIMMR; Parnell, 2019) were used for the two tracer isotopes (13C, 15N). For July 2018 samples, the mixing model was parameterized using the EAM detritus as the mixture and 11 sources: algae tissue (combined Dictyota and Lobophora), BCM tissue, herbivore feces (combined A. bahianus and A. coeruleus), spongivore feces (H. ciliaris), emergent sponge tissue from three species (Aplysina cauliformis, Niphates digitalis, and Xestospongia muta), cryptic sponge tissue from three species (Chondrilla sp, Halisarca sp., Scopalina ruetzleri), and sediment trap samples. The algae tissue types were initially pooled as a single source because of the large overlap between the two species in isotopic space, and the herbivore feces were combined for the same reason. However, sponge species were retained as separate sources in the mixing model due to little overlap in isotopic space. For March 2019 samples, two mixing models were parameterized (one each for EAM and tray detritus) with the same number and type of sources as the 2018 samples, with the addition of Halimeda sp. to the combined algae tissue category and feces of P. paru to the combined spongivore feces category. Each model was run with 10,000 iterations, thinned by 10, and with an initial discard of the first 1,000 iterations. We examined correlations among the posterior distributions in each model to assess the ability of the model to distinguish between different sources, pooled highly correlated sources, and re-ran the model with the pooled sources, keeping other settings the same.

For genetic sequencing data, exact sequence variant (ESV) identifications were tabulated across sample types and years to determine the frequency of occurrence at the level of taxonomic phyla and species, where frequency of occurrence was calculated as the number of samples in which the taxa (either phyla or species) was present, divided by the total number of sequenced samples of that sample type. Multivariate analyses were used to compare the detritus (EAM and tray) from 2018 and 2019. A non-metric multidimensional scaling (NMDS) biplot based on the Jaccard similarity measure was used to visualize differences among detritus types. ESV identifications were converted to presence/absence values at the level of taxonomic order. Ellipses generated from the standard error of the weighted average of the scores in each treatment habitat were superimposed on each biplot to visualize the dispersion of each detritus type. A permutational analysis of variance (PERMANOVA) was conducted to test the multivariate response of taxonomic order presence/absence across the detritus types. When PERMANOVAs were significant (p < 0.05) a similarity percentage analysis (SIMPER) was used to evaluate the contribution of each taxonomic order to differences between detritus types. All statistical analyses were performed in R (R Core Team, 2020).

Results

Stable isotope analyses

Detritus δ13C values ranged from −19.5‰ (2019 EAM detritus) to −14.4‰ (2019 tray detritus), and source δ13C values ranged from −27.7‰ (2019 BCM tissue) to −10.7‰ (2019 Halisarca caerulea tissue; Table S3). Detritus δ15N values ranged from 0.2‰ to 2.7‰ (both from 2019 EAM detritus), and source δ15N values ranged from −2.6‰ (2018 sediment traps) to 5.5‰ (2018 Scopalina ruetzleri tissue). In general, δ15N values were greater in sponge tissue samples than algae and BCM tissue samples (Fig. 1; Table S4). The isotope signatures of the feces of spongivorous fishes (Holacanthus ciliaris, Pomacanthus paru) were similar to sponge tissues, and the isotope signatures of the feces of herbivorous fishes (Acanthurus bahianus, Acanthurus coeruleus) were similar to algae tissues (Fig. 1; Tables S3–S4).

Figure 1 Isotope biplots of samples collected in (A) July 2018 and (B) March 2019.

Detritus samples, including EAM and tray detritus, are represented by open and filled circles, respectively, that denote the values of individual samples. All other symbols represent the average ± SD of source samples.

There were notable seasonal changes in stable isotope compositions of some composite and tissue samples between July 2018 and March 2019. For the δ15N values of sediment trap samples, there was a large increase and shift from negative to positive δ15N values from 2018 (mean ±SD = −1.1 ±1.3‰) to 2019 (−1.7 ±0.5‰; Table S4). The emergent sponge species A. cauliformis and cryptic sponge species Chondrilla sp. exhibited similar isotopic values, and both exhibited increasing δ13C values and decreasing δ15N values from 2018 to 2019 (Fig. 1; Tables S3–S4). The δ13C of the cryptic sponge species Halisarca caerulea also increased from −17.9 ±1.4‰to −12.6 ±2.8‰, and the δ13C of BCM decreased markedly from −21.1 ±0.1‰to −27.1 ±0.5 ‰, and this was the only tissue sample type with reduced δ13C values in 2019 (Table S3).

The Bayesian isotope mixing model based on δ13C and δ15N values was used to establish the relative contribution of each source to the detritus samples (Fig. 1). For 2018 samples, algae and herbivore feces were pooled after initial runs indicated a high correlation between the sample types (correlation = −0.72). The resulting model showed further correlations between the combined algae/herbivore feces source category and BCM tissue and sediment trap samples (correlation = −0.63, −0.74 respectively); these were further pooled into a category “algae/BCM/herbivore feces/sediment trap” (Fig. S1). The resulting model indicated high contribution by the pooled four source mixture to the EAM detritus (mean ±SD = 58.3 ±11.1%), with the other source categories showing much lower contribution (<8.5%; Fig. 2A). Earlier iterations of the model with unpooled sources showed high contribution by algal tissue and herbivore feces (44.8 ±17%, 11.9 ±14.5%, respectively; Fig. S2), suggesting the disproportionate contribution was not merely a byproduct of the pooling of four sources.

Figure 2 Simulated contributions of each source to EAM detritus in (A) July 2018 and (B) March 2019, and to (C) tray detritus in March 2019.

In the box plots, the boundary of the box closest to zero indicates the 25th percentile, a black line within the box marks the median, and the boundary of the box farthest from zero indicates the 75th percentile. Whiskers above and below the box indicate the 10th and 90th percentiles, and outermost points indicate outliers.

The mixing model of 2019 EAM detritus samples showed no major correlations among detrital sources, thus analyses proceeded on the original model with unpooled sources. The mixing model revealed similarly low contributions with large confidence intervals for all sources, ranging from 4.5 ±3.3% for BCM tissue to 16 ±13.5% for algal tissue (Fig. 2B), and likely owing to the greater variability in EAM detritus isotopic values and tighter clustering of source samples around the EAM detritus (Fig. 1B). The tray detritus mixing model showed similarly low contributions and large confidence intervals for all sources (Fig. 2C).

Metabarcoding analyses

A total of 119,012,810 sequence reads were generated with a mean of 577,732 (±302,189 SD) DNA sequence reads retained per sample per marker after bioinformatic filtering. The 18S marker yielded a mean of 1,121 (±2,991 SD) ESVs per sample, COI yielded a mean of 452 (±390 SD) ESVs per sample, and 16S yielded a mean of 11,657 (±26,518 SD) ESVs per sample. Approximately 37.5% of ESVs (295,497 ESVs) were assigned taxonomy based on the selection criteria outlined in the methods.

Genetic sequences of tissue samples largely matched the morphology-based taxonomy (Table S5). Three out of four sponge species analyzed were identified to the genus level, and the fourth sponge species (Halisarca caerulea) was identified at the family level (Halisarcidae), with the expected genus Halisarca scoring a 96%, just below the 98% threshold. The BCM was assigned the expected species Moorea producens, and Lobophora was assigned its proper genus. The tissue samples from the alga visually identified as Dictyota matched to multiple taxa, including phaeophyte genus Dictyopteris and rhodophytes in family Rhodomelaceae including Lomentaria and Neosiphonia.

The composite samples (EAM and tray detritus, as well as source samples from sediment traps and the feces of herbivores and spongivores) contained rich genetic diversity. For the following analyses, we retained eukaryotes and cyanobacteria and excluded Archaea and bacteria other than cyanobacteria. A total of 17 metazoan phyla and 20 phyla from other kingdoms were identified in the composite samples (Fig. 3). Of these phyla, 34 were found in detritus (four unique), 27 were found in sediment trap samples (0 unique), 23 were found in herbivore feces (0 unique) and 21 were found in spongivore feces (2 unique; Fig. 3). Metazoan phyla Arthropoda, Nematoda, Xenacoelomorpha, Annelida, Mollusca, and Porifera were found in over 50% of EAM detritus samples. The most frequent phylum, Arthropoda, was detected in 82% of EAM detritus samples, while Porifera was detected in 55% of EAM detritus samples. Metazoan phyla Arthropoda, Porifera, Chordata, Cnidaria, and Mollusca were found in over 50% of sediment trap samples, and the most frequent phyla were Porifera and Arthropoda (83% each). Metazoan phyla Chordata, Cnidaria, Porifera, and Platyhelminthes were found in over 50% of spongivore feces samples, and the most frequent phyla were Chordata and Cnidaria (80% each) while Porifera was detected in 60% of samples. No metazoan phyla were found in more than 50% of herbivore feces samples (Fig. 3).

Figure 3 The frequencies of occurrence for all phyla in each composite sample in each year.

Number of replicates is shown in parentheses. J18, July 2018; M19, March 2019; EAM, detritus suctioned from epilithic algal matrix; tray, detritus sampled from trays placed under reef overhangs; sed. trap, particulates captured in sediment traps; h’vore feces, feces collected from herbivorous fishes; s’vore feces, feces collected from spongivorous fishes.

On average, non-metazoan phyla had higher frequencies of occurrence than metazoan phyla (Fig. 3). Bacillariophyta, Rhodophyta, Cyanobacteria, and Ciliophora were detected in at least 90% of EAM detritus samples, and those four phyla as well as Chlorophyta, Cercozoa, and Haptista were detected in at least 90% of sediment trap samples. Bacillariophyta, Cyanobacteria and Rhodophyta were detected in at least 90% of herbivore feces samples, and Chlorophyta and Streptophyta were detected in 58% and 50% of herbivore feces samples, respectively. Streptophyta was detected in 100% of spongivore feces samples (Fig. 3).

The main taxa of interest in investigating the sponge-loop hypothesis were sponges, algae, and cyanobacteria. In total, we identified 36 sponge taxa spanning 14 families, 13 genera, and six species (Fig. 4). Multiple sponge taxa were detected in detritus, sediment trap, and spongivore feces samples, including all the sponge taxa identified from the tissue samples, except the family Halisarchidae. However, sponge taxa had very low frequencies of occurrence compared to other algae and metazoans (Fig. 4), and only two of the 36 taxa were detected in more than one of the 11 EAM detritus samples.

Figure 4 The frequencies of occurrence in composite sample types for the 15 most abundant algae and non-sponge metazoans, and all 36 identified sponge taxa.

Number of replicates is shown in parentheses. J18, July 2018; M19, March 2019; EAM, detritus suctioned from epilithic algal matrix; tray, detritus sampled from trays placed under reef overhangs; sed. trap, particulates captured in sediment traps; h’vore feces, feces collected from herbivorous fishes; s’vore feces, feces collected from spongivorous fishes. Classification includes phylum (P), class (C), order (O), family (F), genus (G), and species (S), where applicable. See Tables S6 and S7 for frequencies of all algae and metazoa taxa, respectively.

Among the algae (including Cyanobacteria), we identified 25 classes, 107 orders, 163 families, 202 genera, and 159 species. The most frequently detected taxa in detritus samples were also among the most frequently detected taxa in sediment trap samples as well as herbivore feces samples but were less frequent in spongivore feces samples (Fig. 4; Table S6). The cyanobacterium in the provided reference tissue sample was only detected in EAM and tray detritus, while the Lobophora algae identified from the tissue samples was detected in all sample types (Table S6). The Dictyota algae tissue did not yield consistent taxonomic identifications and the best algal matches from two samples (family Rhodomelaceae and genus Dictyopteris) were detected in EAM detritus, sediment trap samples, and herbivore feces (Table S6).

Metazoans other than Porifera also contributed to detritus. The most frequent metazoan taxa were harpacticoid copepods (phylum Arthropoda), identified in all the EAM detritus samples collected in July 2018 and half of the EAM detritus samples collected in March 2019. Cyclopoid copepods were also well represented in the EAM detritus collected in both years (Fig. 4). Among macrofaunal metazoans, the most frequently occurring in EAM included cnidarians of class Anthozoa (octocorals; Fig. 4), while cnidarians of class Scleractinia (hard corals) were notably scarce, with only one identification in a single EAM sample collected in 2018 (Table S7). Meiofaunal and epifaunal metazoans were more frequent than macrofauna, and Xenacoelomorpha (class Acoela), Platyhelminthes, and Nematoda were frequent in EAM detritus collected in both years (Fig. 4).

There were some notable differences among 2018 and 2019 EAM and tray detritus. The most immediate difference was the number of detected ESVs, and the number of detected ESVs in 2018 EAM detritus (7135 ±8220 ESVs per sample) outnumbered tray detritus (4150 ±4164 ESVs per sample) and greatly outnumbered 2019 EAM detritus (466 ±435 ESVs per sample). There was also relatively low overlap of individual ESVs across seasons, only 245 of the total 34,226 ESVs detected in EAM detritus were present in at least one sample from each season.

The NMDS based on presence-absence of taxonomic orders revealed separation of 2018 and 2019 EAM detritus (Fig. S3). Interestingly, tray detritus collected in 2019 was more similar to 2018 EAM detritus than 2019 EAM detritus, but this may be due to greater similarity in the number of detected ESVs in tray and 2018 EAM detritus. PERMANOVA revealed significant differences in the communities (PERMANOVA F2, 12 = 1.7636, p = 0.029), and pairwise PERMANOVA indicated significant differences between the 2018 and 2019 EAM detritus (adjusted p = 0.024). The phylum with the greatest contribution to dissimilarity was Bacillariophyta (SIMPER: 11.1%).

Frequency of occurrence data revealed some additional potential drivers of dissimilarity between the 2018 and 2019 EAM detritus samples. For the algae and cyanobacterial taxa, 41 of the top 50 most frequent algae taxa were present in 100% of 2018 EAM detritus samples, while only three of the top 50 taxa were present in 100% of 2019 EAM detritus samples (Table S6). Additionally, sequences matching to collected algae tissues were more frequent in 2018 EAM detritus compared to 2019 EAM detritus, particularly Lobophora (Table S6).

Discussion

Unlike studies that led to the proposal of the sponge loop hypothesis (De Goeij et al., 2013) and those that subsequently tested it (McMurray et al., 2018, reviewed in Pawlik & McMurray, 2020) by examining sponge detritus production on the “source side”, this study tested the “sink side” of the sponge loop, exploring the sponge signature in the accumulating detritus on the reef surface. The detritus in the epilithic algal matrix (EAM) on the Carrie Bow Cay forereef was derived from numerous benthic and pelagic sources. Primary producers were the dominant group of organisms contributing to detritus, consistent with previous studies (Hatcher, 1983; Alongi, 1988; Wilson, Burns & Codi, 2001; Wilson et al., 2003). There was genetic signature of sponges (phylum Porifera) in detritus, confirming the validity of the eDNA methodology, but detritus contained low frequencies of Porifera sequences compared to primary producers and other metazoans, and detritus isotopic signatures were distinct from isotopic signatures of sponge tissues, indicating minimal sponge inputs to detritus. The use of complementary analyses and seasonal sampling herein yielded additional insights into how physical conditions (flow, currents, turbidity) may influence detritus composition.

Minimal sponge inputs to detritus

The results of this study are at odds with the detrital component of the sponge loop hypothesis, which proposed that “sponges transform a majority of DOM into particulate detritus” as shed sponge cells with “DOM turnover by sponges approaching the daily gross primary production of the entire reef ecosystem” (De Goeij et al., 2013). Detritus contributions by sponges were proposed to be approximately 6-fold greater than contributions from other sources (De Goeij et al., 2013). Considering these estimates and that the expected form of sponge-derived detrital material is cellular debris, the sponge signal from eDNA and stable isotopic signatures of forereef detritus samples should have been substantial, but it was not.

The six sponges sampled here are common species that represented a cross section of sponge diversity on Caribbean reefs (Loh & Pawlik, 2014). This diversity was evident in the range of isotopic space that species occupied, and such isotopic differences can be attributed to species-specific diets, microbiomes, and nitrogen cycling and fractionation pathways (Freeman et al., 2021; Van Duyl, Mueller & Meesters, 2018). Despite a diverse sampling of sponge species, isotope mixing models indicated low contribution by any species, consistent with the much lower frequencies of occurrence of Porifera sequences in eDNA analyses compared to algae and metazoans, especially when higher-order taxonomic groupings are compared. While frequencies are not true measurements of relative abundance of these diverse organisms, previous studies have demonstrated that DNA frequencies can approximate community abundance (Yoccoz et al., 2012), and low frequencies of Porifera sequences also paralleled differences between sponge and detritus isotopic signatures.

The deployment of trays under reef overhangs in 2019 was a generous test of the sponge loop hypothesis designed to further maximize the odds of capturing choanocytes shed by cryptic sponge species. The three cryptic sponge species targeted in the present study (Halisarca caerulea, Chondrilla caribensis, and Scopalina ruetzleri) were also the same species for which detritus production had been measured using incubation experiments and cited to support the sponge loop hypothesis, with H. caerulea being the most frequently used target species for these studies (De Goeij et al., 2008a; De Goeij et al., 2008b; De Goeij et al., 2009; De Goeij et al., 2013; Lesser et al., 2020; Campana et al., 2021; Hudspith et al., 2021). Notably, none of the EAM or tray detritus samples collected in the present study contained sequences matching to H. caerulea (or Halisarca), despite the relative abundance of this species on the Carrie Bow Cay forereef. Sequences matching to S. ruetzleri and Chondrilla sp. were also absent from all detritus samples, except for one identification of Chondrilla sp. in one of the two sequenced tray detritus samples. The absence of these species in the EAM, where detritivore grazing occurs (Hatcher, 1983), calls into question the ecological relevance of any detritus they may produce.

Detritus inputs by algae and non-sponge metazoans

Isotope mixing models revealed a significant contribution of benthic macroalgae to EAM detritus collected in 2018, and genetic analyses revealed a high frequency of sequences from major plant phyla. These findings are congruent with reports that benthic and filamentous algae are important contributors to coral reef detritus (Wilson, Burns & Codi, 2001). The benthic cover of Dictyota experienced historical increases at Carrie Bow Cay following die-offs of Diadema antillarum in the 1980s and persistent overfishing, but a ban on the harvest of herbivorous fishes in Belize has led to the recovery of herbivore fish populations in recent years (Cox et al., 2013; Mumby & Steneck, 2018; De Pablo et al., 2021). Herbivores not only benefit the reef by minimizing algae cover, but their ceaseless grazing and frequent defecations act as a pathway to recycle algae tissue to the benthos as detritus (Wernberg et al., 2006), and fish egestion of organic matter contain more nutrients than their excretion of inorganic matter (Schiettekatte et al., 2023). A similar function is likely performed by spongivore grazers. Indeed, the feces of prominent spongivore species (Holacanthus ciliaris and Pomacanthus paru, Randall & Hartman, 1968) matched isotopic values of sponge tissues and contained Porifera sequences, validating the use of eDNA methods in attempting to find the sponge signature in reef detritus. However, spongivore feces were isotopically and genetically distinct from EAM detritus, suggesting minimal detritus contributions through the spongivore grazing pathway.

Pelagic sources were well represented in EAM detritus, consistent with the pelagic signature found in detritus collected from the forereef of Palmyra Atoll (Max et al., 2013). All EAM detritus samples contained Bacillariales and Naviculales, orders of diatoms (phylum Bacillariophyceae) known for being among the most abundant microalgal components of detritus, accounting for up to 14% of detrital organic material, and likely contributing important dietary resources such as fatty acids and proteins (reviewed in Wilson et al., 2003). Many EAM detritus samples also contained crustacean signatures at lower overall frequencies, which may represent zooplankton or their feces that have arrived at the EAM through sinking or grazing by the “wall of mouths” (Hamner et al., 1988). Meiofaunal and epifaunal sequences were also abundant in the detritus, likely from living inhabitants of the EAM that are not part of the detritus but may consume it. Sequences matching to copepod order Harpacticoida were notably abundant in EAM detritus, consistent with past observations that harpacticoid copepod nauplii were the most numerically dominant members of the zooplankton on the Carrie Bow Cay forereef (Fornshell, 1994). The xenacoelomorph order Acoela was also found in 73% of detritus samples, and a diversity of these meiofaunal flatworms are known to inhabit the shallow sediments of Carrie Bow Cay (Achatz, Hooge & Tyler, 2007; Hooge & Tyler, 2008). Nematoda were also abundant, but are unlikely to interfere or compete with detritivore grazers (Leduc & Probert, 2009).

The most common macrofaunal metazoans in EAM detritus were Mollusca (55% frequency) and Cnidaria (45% frequency). Identified molluscs included cephalaspidean gastropods in the family Haminoeidae, which may have been a non-selective detritivore grazer akin to other species in this family (Malaquias et al., 2004). Scleractinia (hard corals) were largely absent from EAM detritus. This was unexpected considering longstanding assertions about the importance of coral mucus to reef detritus (Gottfried & Roman, 1983), but missing identifications may also be due to the lack of cellular material containing the targeted genetic sequences in coral mucus. Interestingly, most cnidarian identifications in detritus belonged to octocorals (Order Alcyonacea). Gorgonian octocorals are abundant on Carrie Bow Cay but have received relatively less research attention (Kupfner Johnson & Hallock, 2020). No reports or studies could be found addressing detrital contributions by octocorals, but further study is merited considering the findings herein of abundance of octocoral sequences in detritus.

Unusually negative δ15N in water column particulates collected in sediment traps

The purpose of sediment traps was to capture the particulates sinking from the water column and distinguish them from particulates originating on the benthos. Lower δ13C values of sediment trap particulates compared to benthic detritus were consistent with previous findings (Max et al., 2013) and in agreement with the pattern of generally lower δ13C in phytoplankton compared to benthic autotrophs (Van Duyl, Mueller & Meesters, 2018). Samples from the Palmyra atoll forereef exhibited comparable δ15N values between sediment trap and EAM detritus, indicating a similar pelagic origin and highlighting the importance of pelagic inputs to forereef detritus (Max et al., 2013). In contrast, the isotopic signatures of Carrie Bow Cay forereef detritus suggest a combination of pelagic and benthic inputs, as δ15N of sediment trap particulates was highly reduced compared to EAM detritus, even taking on an unusually negative value in 2018. The observation of lower δ15N in sediment traps (pelagic particulates) compared to EAM detritus (benthic particulates) contradicts other reports of elevated δ15N in pelagic compared to benthic organic matter (Van Duyl, Mueller & Meesters, 2018). This trend reversal may be due to unique circumstances driving negative δ15N in waters around Carrie Bow Cay and reflected in the sediment trap samples in 2018, and two possible explanations for negative δ15N are presented below.

First, uncharacteristically negative δ15N values for detritus in sediment traps from July 2018 samples may be a result of grazing by zooplankton on the diazotrophic cyanobacterium Trichodesmium. Similarly negative δ15N values (−1 to −2‰) of zooplankton collected from oligotrophic waters were attributed to nitrogen fixation by Trichodesmium that are consumed by zooplankton (Montoya, Carpenter & Capone, 2002). Trichodesmium is abundant during the summer at Carrie Bow Cay, and the forereef may act as a sink of Trichodesmium (Villareal, 1995). We identified Trichodesmium in two EAM detritus samples from 2018 (40% frequency of occurrence) and zero sediment trap samples; Trichodesmium colonies were likely either retained in the 150 µm prefilter or grazed, for example by cyanobacteria-grazing mixotrophic dinoflagellates such as Alexandrium (Jeong et al., 2010). Alexandrium was found in 80% of 2018 sediment traps but missing from all other composite samples. This dinoflagellate could have contributed to uniquely negative δ15N values of organic matter if it arrived in large quantities after grazing on Trichodesmium. We were unable to measure the relative abundance of Alexandrium, but this explanation is consistent with reports of sharply declining abundances of Trichodesmium across the Carrie Bow Cay forereef for which zooplankton grazing was implicated as a likely cause (Villareal, 1995).

A second explanation could be that nitrogen in the atmosphere and rain is notably depleted in 15N around Carrie Bow Cay, and the δ15N values of nearby nutrient-starved mangroves can be as low as −17 (Fogel et al., 2008). Currents may carry particulates from the 15N-depleted mangroves from the lagoon, through the tidal cut, and onto the reef. However, July is marked by northeasterly trade winds and a net onshore current flow (Greer & Kjerfve, 1982; Koltes & Opishinski, 2009) that flushes the forereef with more oceanic than lagoonal seawater. Irrespective of the origin of 15N-depleted particles in sediment traps, whether open ocean or lagoon, the isotopic and genetic distinctions between sediment trap particulates and EAM detritus indicate separate origins and composition of the particulates in July 2018.

Seasonal differences in sources and composition of detritus

There were considerable seasonable shifts in the isotopic signatures of several detritus sources between July 2018 and March 2019. Tissue samples of the HMA sponge species Aplysina cauliformis and Chondrilla sp. exhibited a rise in δ13C, perhaps because higher irradiance and water clarity led to more CO2 fixation by the sponge photosymbionts and consequently lower δ13C of these sponge species in July 2018, compared to March 2019 when reduced irradiance may have favored heterotrophy. The δ13C shifts in tissues of other sponge species reflected their different feeding strategies and dietary preferences. The overlap in δ13C between EAM detritus and the tissues of Niphates digitalis is consistent with the detritus-dominated diet of this species (Freeman et al., 2021). The DOC-dominated diets of X. muta and H. caerulea (De Goeij et al., 2008a; McMurray et al., 2018) may also explain the seasonal rise in δ13C values in their tissues, for example if there was a shift from pelagic to benthic sources of DOC; X. muta consumes DOC of varying pelagic and benthic origins, depending on environmental availability (Van Duyl, Mueller & Meesters, 2018).

There was a collective decline in the δ15N values of tissues of all six sponge species in March 2019, suggesting widespread changes to nitrogen taken up by sponges from their environment. The sponge microbiome can also influence δ15N in complex ways (Van Duyl, Mueller & Meesters, 2018) and may explain why some species showed larger δ15N reductions than others. For example, the connection between nitrogen fixation and reduced δ15N (Southwell et al., 2008) may explain larger δ15N reductions in A. cauliformis, which relies on symbionts for nutrition, compared to the smaller δ15N reductions in N. digitalis, which predominantly depends on external nitrogen sources (Weisz et al., 2007).

Samples of tissues from benthic cyanobacterial mats (BCM) exhibited an unexpected drop in δ13C values. This is the first known report of such a seasonal shift for BCM, and it may be due to seasonal variation in dissolved inorganic carbon (DIC) or BCM species composition. Sequencing was only performed on BCM samples from July 2018, precluding seasonal comparison of the species known to occur in the BCM consortia (e.g., Moorea, Oscillatoria, Hydrocoleum).

Although isotopic and genetic signatures of detritus sources varied from July 2018 to March 2019, isotopic signatures of the detritus itself remained relatively consistent between the seasons. EAM detritus contained an abundance of sequences matching to primary producers whose mechanical breakdown and predation can directly contribute to detritus (Wilson et al., 2003). The abundant dissolved organic matter (DOM) produced by benthic algae and other autotrophs may also spontaneously form polymer gels or adsorb to carbonate sediments and become a form of amorphic detritus (Otsuki & Wetzel, 1973; Chin, Orellana & Verdugo, 1998; Wilson et al., 2003). Amorphic detritus is a non-cellular composite undetectable by genetic sequencing methods but with an isotopic signature reflective of its source organism, and significant inputs of this form of detritus may explain similar isotopic signatures despite distinct genetic sequences in detritus from both seasons. Amorphic detritus is a form of molecularly uncharacterized materials, byproducts of complex biogeochemical processes that escape characterization by traditional chromatography-based techniques and are poorly understood across all aquatic ecosystems, especially coral reefs (Bowen, 1979; Wilson, 2002; Wilson et al., 2003; Wakeham & Lee, 2019).

Multiple lines of evidence indicate that detritus composition was shaped by environmental conditions on the forereef, namely higher flow and greater turbulence in March 2019 compared to July 2018. First, detritus collected in March 2019 had more variable isotopic delta values and fewer genetic sequences compared to detritus collected in July 2018, indicating a more heterogenous and scarce mixture that may result from greater turbulence and mixing. Detritus collected in March 2019 was also characterized by greater frequency of meiofauna and epifauna that likely inhabit the EAM, including Annelida, Nematoda, and Xenacoelomorpha, and lower frequencies of macrofauna that originated elsewhere on the benthos, pointing to surge-driven export of macrofaunal-derived organic matter before it could settle and get trapped in the EAM in March 2019. Stronger surge conditions may have also caused the resuspension of benthic particulates in the water column, which would account for the presence of sequences belonging to benthic metazoans (e.g., Porifera, Mollusca, Cnidaria) in sediment trap samples in March 2019. Sediment traps are effective water column samplers when conditions are calm (flow < 0.2 m/s), as was the case during July 2018 sampling, but sediment traps may capture resuspended benthic materials when conditions are turbulent as was the case during March 2019 sampling (Gardner, 1980; White, 1990; Wilson et al., 2003). The more energetic environment in March 2019 may have also amplified production of amorphic detritus in the EAM, as DOM aggregation may be facilitated by more encounters with substrates and increased laminar shear and bubbles formed in the higher water motion (Wilson et al., 2003).

Ecological implications

Organic matter cycling through the detrital food web is complicated by the number of organisms that may act as both sources and sinks of detritus, including bacteria, sponges, surgeonfishes, and parrotfishes (Choat, Clements & Robbins, 2002; Wilson et al., 2003; Crossman, Choat & Clements, 2005; Dromard et al., 2015; Mumby & Steneck, 2018). It is important to understand how detritus is processed and transformed by these groups. Certain detritivores can selectively reduce protein levels on the reef, resulting in higher carbon-to-nitrogen ratios and decreased nutritional value in nitrogen-limited systems, as observed in the backreef habitats of Palmyra Atoll in the Pacific Ocean (Max et al., 2013). Detritus is likely more abundant on Caribbean coral reefs (Mumby & Steneck, 2018), and there may be a surplus of nutritious detritus on the Carrie Bow Cay forereef due to the windward exposure of the reef and other favorable conditions driving import of particulate matter onto the reef.

The higher productivity in the Summer (July 2018) and potential influx of sinking phyto- and zooplankton may have supplemented benthic sources of detritus, while the more energetic early spring (March 2019) resulted in greater mixing and flushing and perhaps better conditions for formation and accumulation of amorphic detritus in the EAM. Seasonal conditions may bring a variety and surplus of detritus that is regularly replenished to the forereef and may even supply downstream backreef and lagoonal habitats (Hatcher, 1983). Despite an abundance of detritus, there may be an imbalance in the number of detritivores feeding on it, as detritivores are less diverse and abundant in the Caribbean compared to Pacific reefs (Bellwood et al., 2004; Roff & Mumby, 2012; Edwards et al., 2014). A shortage of detritivores in the Caribbean may leave this overflow detritus to get recycled into the microbial loop, returned to benthic suspension feeders (e.g., sponges), or lost from the reef system (Mumby & Steneck, 2018).

It may be argued that this “sink side” test of the sponge loop is based on a single coral reef location, but as indicated above, the location was chosen to maximize the likelihood of confirming the hypothesis, with among the highest sponge abundances reported for Caribbean coral reefs (Wilkinson & Cheshire, 1990; Loh & Pawlik, 2014) and the deployment of trays directly under reef overhangs that supported cryptic sponge species. Despite the foregoing, the sponge signature in reef detritus was negligible. The breadth of sampling conducted in the present study (16 tissue and composite sample types collected over two seasons) encompassed many common organisms and the most likely candidates for detritus production, although this sample breadth limited the number of replicates from each group that could feasibly be analyzed. The stable isotope analyses and eDNA metabarcoding analysis methods used herein each had limitations that the other method accounted for to some degree, but some gaps remained. Isotope mixing models can only estimate contributions of sampled sources, so while this was an expansive sampling effort, it was not exhaustive, and detritus isotopic signatures may have been influenced by diagenetic changes, especially of δ15N values (Caraco et al., 1998). Genetic sequencing complemented the isotopic analyses by identifying all traces of organisms that are cataloged and with genetic material that resisted degradation, but some organisms could have escaped detection if they were novel or degraded (Reef et al., 2017). The omission of bacteria besides cyanobacteria in the genetic sequence data also discounted a key component of detritus (Biddanda, 1985; Alongi, 1988; Wilson et al., 2003), but the bacterial signal would have been present in the isotopic signatures of detritus.

Conclusion

To better understand how reefs of the future will function, it is crucial to understand how energy is transferred throughout contemporary coral reefs, including the complex and poorly-elucidated detrital pathway. While sponges may not be major contributors through direct production of detritus, they may extend the residence time of organic matter on coral reefs through efficient uptake of DOC and accumulation of biomass that is eventually grazed or mechanically broken down and returned to the detrital pool. This carbon pathway may not be essential to the healthy functioning of Caribbean coral reefs, however, considering low demand by relatively few detritivorous grazers. As benthic communities continue to shift from coral- to algae- or sponge-dominated systems, this will alter trophodynamics and influence the detrital resources on which these systems rely.

Supplemental Information

Supplemental Information 1 Supplemental Information about Environmental DNA Sample Laboratory Processing

Figure S1 Isotope biplot of samples collected in July 2018, with combined source category of algae tissue, BCM tissue, herbivore feces, and sediment trap

EAM detritus samples are represented by open circles, respectively, that denote the values of individual samples. All other symbols represent the average ± SD of source samples denoted in legend.

Figure S2 The contribution of each source to EAM detritus July 2018, before combining algae tissue, BCM tissue, herbivore feces, and sediment trap

In the box plots, the boundary of the box closest to zero indicates the 25th percentile, a black line within the box marks the median, and the boundary of the box farthest from zero indicates the 75th percentile. Whiskers above and below the box indicate the 10th and 90th percentiles. Points above and below the whiskers indicate outliers outside the 10th and 90th percentiles.

Figure S3 NMDS of the taxonomic order presence/absence values in each detritus sample

Points represent individual samples, and 95% confidence ellipses indicate sampling distributions for each detritus type.

Table S1 Primer set used to amplify each gene region of each sample

Supplemental Information 6 DNA Markers amplified from each sample

Table S3 Descriptive statistics for δ13C of detritus and detritus sources collected in July 2018 and March 2019

Composite samples are indicated in bold.

Table S4 Descriptive statistics for δ15N of detritus and detritus sources collected in July 2018 and March 2019

Composite samples are indicated in bold.

Table S5 Best taxonomic matches for tissue samples

For each taxonomic level, the threshold similarity score used for assignment is listed. Empty cells indicate that the taxonomy for that sample at that level did not meet the threshold or was ambiguous (i.e., matched two groups equally).

Table S6 The frequency of occurrence of all algae and cyanobacterial taxa in the composite sample types that were present in at least one EAM detritus sample

Classification includes phylum (P), class (C), order (O), family (F), genus (G), and species (S), where applicable. EAM = detritus suctioned from epilithic algal matrix in July 2018 (J18; n = 5) and March 2019 (M19, n = 6). sed. trap = particulates captured in sediment traps in July 2018 (n = 5) and March 2019 ( n = 7). tray = detritus sampled from trays placed under reef overhangs in March 2019 (n = 2). h’vore fec = feces collected from herbivorous fishes in 2018 (n = 5) and 2019 (n = 7). s’vore fec = feces collected from spongivore fishes in 2018 ( n = 1) and 2019 (n = 4).

Table S7 The frequency of occurrence of all metazoan taxa (excluding Porifera) in the composite sample types that were present in at least one EAM detritus sample

Classification includes phylum (P), class (C), order (O), family (F), genus (G), and species (S), where applicable. EAM = detritus suctioned from epilithic algal matrix in July 2018 (J18; n = 5) and March 2019 (M19, n = 6). sed. trap = particulates captured in sediment traps in July 2018 (n = 5) and March 2019 (n = 7). tray = detritus sampled from trays placed under reef overhangs in March 2019 (n = 2). h’vore fec = feces collected from herbivorous fishes in 2018 (n = 5) and 2019 (n = 7). s’vore fec = feces collected from spongivore fishes in 2018 (n = 1) and 2019 (n = 4).

Data S1 All data and R markdown code used for analyses

Raw stable isotope measurements are available in SIAdata_30sept2023.xlsx, and code for SIA analyses are available in files starting with “SIMMR_*.Rmd”. The raw sequence tables from metabarcoding are available in .csv files starting with “edna_CO1”, “edna_18S”, and “edna_16S”, and the code to analyze sequence data are in .Rmd files starting with “edna_1_analysis”, “edna_2_analysis”, and “edna_3_analysis”. The pdfs in “markdown_outputs/” subdirectory are retained for quick reference and reproducible by knitting the .Rmds.

Many thanks to Laura Gaitan Daza, Steve McMurray, Jim Evans, Mellissa Dionesotes, and Sasha Giametti for assistance in the field, the staff of Smithsonian Institution’s Carrie Bow Cay Field Station in Belize for logistical support, Evan Heit and Kim Rosov for assistance with preparation and analyses of stable isotope samples, and Nicole Fogarty and Amy Grogan for additional assistance with reviewing and providing input on earlier drafts of this manuscript.

Additional Information and Declarations

Competing Interests

Author Contributions

Field Study Permissions

DNA Deposition

Data Availability

Beverly McClenaghan, Lesley Berghuis, Nicole Fahner and Hoda Rajabi are employees of eDNAtec Inc. Mehrdad Hajibabaei is the founder and Chief Scientific Officer of eDNAtec Inc.

Lauren K. Olinger conceived and designed the experiments, performed the experiments, analyzed the data, prepared figures and/or tables, authored or reviewed drafts of the article, and approved the final draft.

Beverly McClenaghan analyzed the data, authored or reviewed drafts of the article, and approved the final draft.

Mehrdad Hajibabaei analyzed the data, authored or reviewed drafts of the article, and approved the final draft.

Nicole Fahner analyzed the data, authored or reviewed drafts of the article, and approved the final draft.

Lesley Berghuis analyzed the data, authored or reviewed drafts of the article, and approved the final draft.

Hoda Rajabi analyzed the data, authored or reviewed drafts of the article, and approved the final draft.

Patrick Erwin conceived and designed the experiments, authored or reviewed drafts of the article, and approved the final draft.

Chad S. Lane conceived and designed the experiments, authored or reviewed drafts of the article, and approved the final draft.

Joseph R. Pawlik conceived and designed the experiments, performed the experiments, authored or reviewed drafts of the article, and approved the final draft.

The following information was supplied relating to field study approvals (i.e., approving body and any reference numbers):

Field experiments were approved by the Belize Fisheries Department (permit number 00023-18).

The following information was supplied regarding the deposition of DNA sequences:

The DNA sequence data generated during and/or analyzed during the current study are available at GenBank: PRJNA965826.

The following information was supplied regarding data availability:

The raw data is available in the Supplemental Files.

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
