# Peer review of "Looking for the sponge loop: analyses of detritus on a Caribbean forereef using stable isotope and eDNA metabarcoding techniques"

_PeerJ, doi:10.7717/peerj.16970_

## Round 0.1 · original submission · Minor Revisions

Both reviewers were very positive about this paper and recognised the step forward that the application of eDNA + SIA techniques represents. However, they raise some valid concerns that need to be addressed. In particular the comments from Reviewer 1 who points out that the difference in method from previous studies may be important with regard to interpretation of the results. The fact that your findings are different to some previous studies may reflect the method as much as anything else. "Coral" reefs around the world are very different as I'm sure you are aware and, in agreement with what Reviewer 2 alludes to, I wonder how much of the controversy around the sponge loop could be to do with the massive differences between Caribbean, Deep sea, Red Sea and Asian systems? I cannot believe the systems are identical with regard to the sponge loop. I do wonder if this might be worth a sentence or two in the discussion? It may be particularly important given the slow death of reefs around the world and the likelihood that any existing coral mucus > sponge > POC cycles may be weakened or altered in the future. Reviewer 2 also suggests that the link between introduction and discussion needs to be tightened up a bit and suggests that you consider changing the title to reflect the discussion more.

Congratulations on such well-presented and annotated code!

I have annotated a pdf with a few comments on it. Generally, I found your paper to be extremely well-written and a pleasure to review.

·

Basic reporting

a. Writing is clear, unambiguous and professional.
b. Professional article structure, figures, tables and raw data is shared.
c. Literature references are included and relevant
d. The background/context clearly lays out the novelty of the research presented in the paper. Some detail could be removed to condense the introduction, but this does not detract from the flow and/or quality of the introduction.
e. Minor Revision necessary:
i. Some comparisons made later in this paper are not included in the introduction. The aim of the paper, as stated in the last paragraph of the introduction, is outlined as an exploration of the, ‘composition of detritus, with a particular focus on sponges to test claims that sponges are major sources of detritus on Caribbean coral reefs’ (line 136-137). However, the analysis and results go further than this and there are several detailed comparisons between the two sampling seasons. Either the scope/aim of the paper needs to be more clearly worded/defined or the results need to be parsed back to focus on the aim that is presented initially.
1. Only some of the focus of the results is aimed at the outcomes of the sponge evaluation.

Experimental design

a. This is original primary research that fits within the aim and scope of the journal.
b. Research question is defined, relevant and meaningful (although the focus on sponges is somewhat questionable throughout). This research fills an identified knowledge gap
c. Investigation is rigorous and performed to a high technical and ethical standard
d. Methods are described with sufficient detail and information to replicate
e. Notes:
i. Authors have curated an elaborate array of sources to evaluate the flow of detritus within a Caribbean coral reef system. The collection and sampling methods are well laid out to paint the picture for how different possible sources and sinks for detritus are sampled and quantified.
f. Minor Revisions/discussion needed:
i. It is unclear if this entire study was conducted at one sample location at one reef, across the two seasons (lines 140-152). Potential issues of pseudoreplication? Would have been good to include more characterisation of the sample locations themselves as well as the local biodiversity/sponge distribution (given the emphasis on sponges in the title, abstract and introduction).
ii. Sample sizes are relatively low (Table 1; several are n=0–2). Some acknowledgement of this limitation needs to be made in the discussion.
iii. Why were the samples collected at varying sampling efforts across different seasons (i.e., almost 3x sediment traps for SIA between July 2018 and March 2019)? The justification/reasoning for the imbalance of sampling effort and the reason for sampling during different seasons needs to be made clear.
iv. No quantification of sponge abundance from the target reef. How does this compare to other reefs where the sponge loop hypothesis was proposed?
v. What is the condition of the reef that was included in this study? (some reference to this is made in the discussion, but how does this reef compare to other reefs where the sponge loop hypothesis was proposed?) Could poor reef condition be contributing to the dominant detrital presence of macroalgae?

Validity of the findings

a. Underlying data have been provided that is robust, statistically sound and controlled.
b. Conclusions are well stated and linked to original research question and limited to supporting results.
c. Notes:
i. Findings are presented in a comprehensive and understandable way.
d. Minor Revisions/discussion needed:
i. Figure 4 shoes the frequencies of occurrence in composite sample types. However, even the s’vore faecal samples show a low frequency/occurrence of sponge species. As alluded to in the comments above: is this due to a low distribution/abundance of sponge species to this particular reef? There needs to be more evaluation of this nature before the hypothesis in question (sponge loop hypothesis) can be refuted.
ii. Similarly, Figure 2 and much of the results/discussion refer to the fact that the majority of EAM detritus is algae tissue. Is this due to poor condition of the reef that was sampled here? (this is alluded to in the discussion but there is no evaluation of hard coral cover vs algal cover given in this paper to help clarify this). Again, this sort of information would be useful before the sponge loop hypothesis can be properly refuted. Perhaps this particular reef is currently in poor condition with low sponge diversity/abundance, which may have contributed to these divergent findings.

Additional comments

a. The paper sets out to be focused on the contribution of sponges to detritus on a Caribbean coral reef system (given the title, abstract and introduction). However, much of the focus is on the overall characterisation of detritus sources and seasonal comparisons/variation in detritus. While the findings that contradict the sponge loop hypothesis are interesting, I would recommend that the title, abstract and introduction be changed to reflect what is being focused on and reported throughout.
i. For example, much of the deliberation in the discussion is focused on the seasonal differences/variation and sources for detritus and not on why sponges may not be contributing as much as would be expected to detritus material that was sampled.
b. Overall, the work is comprehensive and enlightening throughout and merits publication and dissemination. However, there are certain elements and gaps that need addressing that have been highlighted above. It is likely the case that these issues have been considered but are not clearly addressed in the manuscript currently and would likely need some tweaking/addressing.

Reviewer 2 ·

Basic reporting

Olinger and colleagues in their manuscript titled "Looking for the sponge loop: analyses of detritus on a Caribbean forereef using stable isotope and eDNA metabarcoding techniques" investigate the composition of detritus in Caribbean coral reefs, specifically testing the hypothesis that cryptic sponges are significant contributors to detritus as part of a hypothesised sponge loop. The study was conducted at Carrie Bow Cay and involved collecting detritus samples from the epilithic algal matrix (EAM) and potential sources of detritus. The researchers utilised stable isotope analyses and environmental DNA (eDNA) metabarcoding to determine the composition of the detritus.

I applaud the authors in their effort to test this hypothesis in their natural experiment set up at Carrie Bow Cay field station. This was a well thought out and thorough investigation to test the sponge loop hypothesis and in generating an immense data set. The strength of this study lies in the dual approach of using stable isotopes and DNA metabarcoding of multiple sources of the composition of detritus in coral reef ecosystems. This combination of methods represents a significant advancement over traditional approaches and contributes valuable insights into the functioning of coral reef ecosystems. That said, there is a notable absence of a detailed comparison between your methodological approach and those used in key studies within this domain, such as de Goeij et al. (2013). Understanding the methodological differences between studies is crucial, as it offers insights into why findings may vary across different research. The scale, sensitivity, and focus of methodologies can significantly influence outcomes. For instance, your eDNA metabarcoding technique allows for the detection of a wide array of organisms, including those not easily identifiable through traditional sampling methods. This could account for differences in the identified sources of detritus compared to studies using more conventional methods. I suggest that the authors provide a concise discussion on how the methodological approach of your study—specifically the use of eDNA metabarcoding—diverges from that of the original sponge loop study. This discussion should be framed in a way that acknowledges the complexity of testing such a hypothesis, given the ecological variability and the fine-scale resolution offered by eDNA metabarcoding. It is important to note that these differences do not necessarily imply any inherent issues with your study (which it sometimes reads) but rather reflect its methodological distinctiveness and sensitivity. Highlighting these distinctions can help to contextualise your findings without diminishing the value of your research or the contributions of the work done.

Methods:

Line 171: why was a 72 hour period used to sample the detrital material?
Line 247-249: Provide a brief overview of what extraction kits are being used for which sample types and then refer the supplemental information for further details.
Line 271-274: revise this statement to simply mention that potential contaminants were removed without stating pests or food as the life cycle of mosquitoes and beetles have an essential aquatic component as larvae.

Experimental design

na

Validity of the findings

na

Reviewer 3 ·

Basic reporting

.

Experimental design

.

Validity of the findings

.

Additional comments

This study tackles an important and contemporary question, namely: how important is the proposed
sponge-loop in the provisioning of detritus to coral reef detritovores and therefore the recycling of
carbon and nitrogen on reefs? The manuscript is well-written, but long and very dense with information
– so it’s not an easy read.
The authors conclude that sponges are not particularly important contributors to detritus on coral reefs
in contradiction to sponge-loop predictions. Yet, the data summarized nicely in Figure 2 seem a bit at
odds with their conclusion. It is clear that the algae-BCM-feces-sediment trap categories dominate when
lumped together (explicitly in July), but the sponge-derived detritus when similarly lumped together also
appears to be a major fraction of the detritus.
One gets lost in the details presented in the Results and Discussion that steer the reader away from the
idea that sponge detritus is important, but I keep coming back to the data illustration in Figure 2 which
doesn’t seem to support such a conclusion. My take is that the composition of reef detritus is
unsurprisingly rich and variable in its sources. Benthic algae comprise a large fraction, pelagic POM is
seasonally important, octocoral detritus is surprisingly common – but the contribution of sponges both
cryptic and emergent is also appreciable. If that conclusion is correct, then the sponge-loop isn’t refuted.
It may not be the primary source of detritus on reefs, but it still appears to be an important component
that was previously not appreciated, much like the octocoral findings highlighted here.
Another means of investigating the detritus-community structure relationship that was not pursued,
presumably due to logistical constraints, is to use similar methods at independent sites that differ in
sponge abundance/diversity. It would be illustrative and relevant to see if sponge biomass (cryptic and
erect sponges) scales with detrital production. Digging even deeper, one might need to quantify the
detrital production of individual sponge species and size and then use those data along with sponge
community abundance and size-structure to estimate the relative contribution of sponges to the detrital
trophic pathway.
From a methodological point of view, the methods used in this study appear solid although I do not
claim exceptional expertise in stable-isotope or eDNA analyses. One potential flaw is the use of
sediment traps to approximate waterborne POM input. Pelagic POM is best determined by filtering a
large volume of oligotrophic water; it’s a standard practice. The authors acknowledge this issue and
offer some explanation and justification as to why sediment traps might serve as an adequate proxy, but
without some correlative data or reference to a study that confirms this, I remain skeptical.
In conclusion, this is a scholarly and worthwhile contribution to the on-going debate over the
importance of the sponge-loop in helping to explain “Darwin’s Paradox” regarding the coral reef
productivity. My only suggestions for alterations in the manuscript are to:
1. Seek to omit some tangential details that muddy the story and make the manuscript difficult to
comprehend, and
2. Rethink without bias whether sponge-derived detritus is truly unimportant, which is the
message that is espoused. How much detrital production is enough before that source is
deemed important for trophodynamics: 5%, 10%, 30%, etc.?

---

## Round 0.2 · accepted · Accept

I still think that Figure 2 would be improved by jittering your points to help indicate the distributions but it's not enough of a gripe for me to hold up the progress of your paper. Well done on a nice piece of work.

One thing I would recommend in future when responding to referee/editor comments is to respond to all of them (even the ones you decide not to action or where the editor/referee might have missed something.